# Regression-based modeling of pairwise genomic linkage data identifies risk factors for healthcare-associated pathogen transmission: application to carbapenem-resistant *Klebsiella pneumoniae* transmission in a long-term care facility

Hannah Steinberg,[1,2] Timileyin Adediran,[3] Mary K. Hayden,[4] Evan Snitkin,[3,5,6] Jon Zelner[1,2]

**ABSTRACT** Pathogen whole-genome sequencing (WGS) has significant potential for improving healthcare-associated infection (HAI) outcomes. However, methods for integrating WGS with epidemiologic data to quantify risks for pathogen spread remain underdeveloped. To identify analytic strategies for conducting WGS-based HAI surveillance in high-burden settings, we modeled patient- and facility-level transmission risks of carbapenem-resistant *Klebsiella pneumoniae* (CRKP) in a long-term acute care hospital (LTACH). Using rectal surveillance data collected over 1 year, we fit three pairwise regression models with three different metrics of genomic relatedness for pairs of case isolates, a proxy for transmission linkage: (i) single-nucleotide variant genomic distance, (ii) closest genomic donor, and (iii) common genomic cluster. To assess the performance of these approaches under real-world conditions defined by passive surveillance, we conducted a sensitivity study including only cases detected by admission surveillance or clinical symptoms. Genomic relatedness between pairs of isolates was associated with room sharing in two of the three models and overlapping stays on a high-acuity unit in all models, echoing previous findings from LTACH settings. In our sensitivity analysis, qualitative findings were robust to the exclusion of cases that would not have been identified with a passive surveillance strategy; however, uncertainty in all estimates also increased markedly. Taken together, our results demonstrate that pairwise regression models combining relevant genomic and epidemiologic data are useful tools for identifying HAI transmission risks.

**IMPORTANCE** Whole-genome sequencing of healthcare-associated infections (HAIs) is becoming more common, and new methods are necessary to integrate these data with epidemiologic risk factors to quantify transmission drivers. We demonstrate how pairwise regression models, in which the outcome of a regression model represents genomic similarity between a pair of isolates, can identify known transmission risk factors of carbapenem-resistant *Klebsiella pneumoniae* in a long-term acute care facility. Such pairwise regression models could be used with rich epidemiologic data in other settings to identify important risk factors of endemic HAI transmission.

**KEYWORDS** carbapenem-resistant *Kleabsiella pneumoniae*, microbial genomics, epidemiology, regression models, healthcare-associated infections, transmission risk factors

Despite intensive research and scrutiny, healthcare-associated infections (HAIs) remain among the most frequent adverse events occurring in health facilities throughout the United States and the world. Improvements in broadly effective infection

Address correspondence to Evan Snitkin, esnitkin@med.umich.edu.

Evan Snitkin and Jon Zelner contributed equally to this article.

The authors declare no conflict of interest.

See the funding table on p. 10.

prevention interventions such as hand hygiene and environmental cleaning, and targeted interventions such as pathogen decolonization, have been attributed to recent reductions in HAIs (1). Still, it is estimated that on any given day 1 in 31 hospital patients in the United States has at least one HAI (1). HAIs are among the top 10 causes of death in the United States and are associated with billions of dollars in excess healthcare costs (2). Antibiotic resistance is common in healthcare pathogens (3, 4) and can make these infections harder to treat.

Colonization typically precedes infection, and many HAI prevention strategies work by interrupting transmission of colonizing pathogens. However, a better understanding of drivers of transmission and of which patients are more likely to transmit or acquire colonization could help in developing more effective interventions to reduce HAIs.

With the increased availability and falling cost of whole-genome sequencing (WGS), there has been increased interest in the use of WGS to understand transmission pathways in healthcare settings (5). However, many studies of transmission in healthcare settings are descriptive in nature, identifying shared exposures among individuals with genomic linkage, but not quantitatively evaluating whether exposures are shared more than would be expected by chance. Thus, understanding risk factors for transmission and identifying putative targets for improved infection prevention will require more rigorous methods that integrate genomic and epidemiologic data to quantitatively identify transmission risk factors. While this has been done to some degree in outbreak settings (6, 7), there is still a need for methods applicable in high-prevalence endemic settings, where the constant importation of resistant organisms and frequent horizontal gene transfer of resistance-conferring genetic elements (8) makes delineating transmission links challenging, even with genomic data.

In addition to the lack of standard frameworks for integrating genomic with epidemiologic data, additional barriers to current methods (e.g., single nucleotide variant, or SNV,-based regression models [9], machine learning algorithms [10], and probabilistic transmission models [11]) include the requirement of SNV cutoffs to infer transmission (9, 10), needing data on uninfected controls (10), and models that are complex (10) and/or require many assumptions about the transmission system hindering their generalizability (11). Additionally, models that do not account for the disproportionate effect of super-spreaders on model outcomes may overestimate confidence in risk-factor estimates (12). Recent work has shown that pairwise models that utilize individual, pairwise, and contextual data to describe the genetic relatedness of pathogen isolates in an endemic setting (12) can identify epidemiologic drivers of transmission with fewer assumptions and computational needs than some previous studies and do not require data on non-cases or SNV cutoffs. This method utilizes WGS data from infected cases and involves a regression model in which the outcome is a measure of genetic similarity between a pair of isolates, and covariates are assessed for their influence on genetic similarity, which can be considered in many cases a proxy for transmission.

In this analysis, we aim to further validate the use of pairwise regression models incorporating WGS and epidemiologic data to study infectious disease transmission in endemic settings by evaluating the use of these models to describe how carbapenem-resistant *Klebsiella pneumoniae* (CRKP), an important healthcare-associated pathogen, is transmitted in a long-term acute care hospital (LTACH) using isolates collected on a regular basis as part of a surveillance study (13). CRKP's high prevalence in LTACHs makes delineating transmission pathways complex with traditional epidemiologic methods. Although we have limited epidemiologic information in this data set, we are able to identify known transmission risk factors with our models, and we hope this study can serve as an example of how to conduct this type of analysis in settings with richer epidemiologic data.

In addition to evaluating different approaches for incorporating genomic relatedness into pairwise statistical models, we also assess the sensitivity of these models to case capture. Sampling strategy may be important in understanding CRKP transmission as

asymptomatic colonization with CRKP is a common precursor to invasive infection (14) and potentially important in intra-facility spread (15), yet most facilities do not screen for asymptomatic colonization. To understand the impact of the sampling scheme on the identification of transmission risk factors, we evaluated models with a more passive surveillance strategy for the detection of carriers.

## MATERIALS AND METHODS

### Study population

CRKP surveillance samples were collected via rectal swab on admission and every 2 weeks from June 2012 to June 2013 for all patients ($n$ = 937 unique patients) in an LTACH in Chicago, Illinois (USA) (13). Average daily patient census was 98 (SD: 7.4), and the median length of stay was 27 days (IQR: 17–44). The mean age of patients was 60.5 years (SD: 15.8), and 43.1% of patient-days were for ventilated patients. This study (HUM00174826) was approved by the institutional review boards at Rush University Medical Center (Chicago, IL, USA) and the University of Michigan (Ann Arbor, MI, USA). Informed consent was waived.

Surveillance samples were cultured, and unique colony morphologies were identified to species level. Ertapenem disks were used to screen isolates for carbapenem resistance, and a confirmatory PCR was conducted to detect $bla_{KPC}$, the sole carbapenemase gene associated with carbapenem-resistant *Enterobacteriaceae* (CRE) in the region during the study period (16). To capture contact patterns, each patient's daily room and floor locations were recorded. Antibiotic usage over time was also recorded for each patient. Whole-genome sequences were obtained for all isolates that were positive on both ertapenem disk and $bla_{KPC}$ PCR assays, and recombination-filtered core genome alignments were produced for each sequence type (17). For this analysis, only sequence type 258 (ST258) isolates, the most common sequence type in the LTACH (70% of cases), were used.

### Regression models of pairwise genomic relatedness

We constructed pairwise regression models in which each observation was a pair of CRKP-positive patients with the individual in the pair who tested positive first being considered the donor and the other individual the recipient. We assessed three different measures of genomic relatedness to be used as outcomes in these models: (i) a log-linear model of core genome SNV distances between the two isolates (raw SNV distances were calculated with the *dist.dna* function in the *ape* R package v5.8-1 [18]), (ii) a logistic regression model with a binary outcome indicating whether the potential donor was the most closely related potential donor for a given recipient (based on SNV distance), and (iii) a logistic regression model with a binary outcome indicating if the pair's isolates were previously determined to be in the same genomic cluster using a threshold-free clustering method (17). Each of these measures assesses the extent to which cases are linked by transmission, with lower genomic distance or cluster co-membership indicating a higher likelihood of direct transmission.

Only donor-recipient pairs where the two individuals overlapped in the facility during their pairwise exposure period (from the last time the potential donor tested negative, or time of admission for admission-positive donors, to the first time the recipient tested positive) were included in the final models. We also ran models with all pairs, including those that did not have overlapping stays in the facility, as a sensitivity analysis. If multiple isolates were available for a patient, only the closest related isolates (based on SNV distances) for each pair of patients were used. To account for the influence of unusually infectious individuals and nonindependence of pairs with shared cases, all models included a random intercept term for each potential donor. Covariates in the models included whether the pair shared time on the same floor (and which floor) or in the same room during their exposure period, time between sample collection of

positive cultures in weeks (a measure of similarity of colonization timing), dichotomous antibiotic receipt by the donor during the exposure period (stratified into carbapenem and non-carbapenem groups), and time period within the study based on when the recipient tested positive (broken up into quarters). Sequential room sharing was also evaluated in our sensitivity analysis that included non-overlapping pairs. Statistical analyses were conducted using *R* v.4.2.2 (19) and all models were run using the *glmer* function in the *lme4* package v1.1-35.3 (20).

### Evaluation of the effect of serial sampling on risk-factor estimates

As most facilities do not have robust serial sampling strategies like our study facility implemented, we re-ran all models including only patients who tested positive on admission or had a positive CRKP test as part of clinical evaluation outside of the colonization study to examine the influence of serial sampling on the ability to make inferences on transmission dynamics.

## RESULTS

### Prevalence of colonization and infection with CRKP in a single LTACH

In total, 255 individuals were colonized with at least one strain of CRKP during the study period (with an average prevalence of 32% throughout the year) (17), 180 of whom (70% of those colonized) were colonized with strain *ST258*. Of the 180 patients colonized with CRKP ST258, 87 (48%) were positive on admission, 72 (40%) had CRKP detected via clinical testing, and 54 (30%) were detected after admission during serial sampling and never had a clinical CRKP isolate. Six patients colonized with *ST258* were excluded from further analyses due to low-quality genome sequences. There were 37 genomic clusters of ST258 (2–16 isolates per cluster) previously identified in this study population with a threshold-free cluster detection approach that clustered each CRKP isolate acquired at the LTACH to the importation isolate with which it shared the greatest number of variants (17).

### Genetic relatedness was greatest among CRKP pairs sharing a room or floor

The median pairwise SNV distance between all *ST258* isolates who overlapped in the facility was 53 (IQR: 38–86); for closest donor pairs and same-cluster pairs, this value was 5 (IQR: 1–24) and 3 (IQR: 1–6), respectively, which are consistent with a previous study identifying 21 SNVs as an appropriate cutoff for *ST258* intra-facility transmission (21) (Table 1; Fig. 1). Room and floor sharing (particularly Floor D which housed the high acuity unit) during the pairwise exposure period was more common among closest-donor pairs and same-cluster pairs than for all possible pairs (Table 1).

Twenty percent (95% CI: 9–38%) of pairs who shared a room during their exposure window had CRKP isolates in the same cluster, and 15% (95% CI: 5–31%) were closest donor pairs. Pairs who shared a floor but not a room during their exposure period were in the same cluster 7% of the time (95% CI: 5–8%) and contained a closest potential donor 5% of the time (95% CI: 4–6%). By contrast, pairs that did not overlap in the same room or floor during their exposure period were in the same genomic cluster only 3% of the time (95% CI: 2–4%) and the closest potential donor to their recipient 3% of the time (95% CI: 2–4%) (Fig. 2). Fifty-eight percent (95% CI: 50–67%) of closest donor pairs were in the same genomic cluster, while only 3% (95% CI: 2–3%) of non-closest donor pairs were in the same cluster.

### Pairwise models suggest room sharing, residing on the floor that housed the high acuity unit, and shorter time lags between colonization detection are associated with genetic relatedness

In all pairwise models, shared time on either Floor A or Floor D was associated with increased pairwise genomic relatedness, with Floor D (which contained the facility's high

**TABLE 1** Individual, dyadic, and contextual characteristics of pairs of CRKP-infected or colonized patients[a]

| | Possible pairs (*n* = 3,500) | Closest donor pairs (*n* = 130) | Same cluster pairs (*n* = 165) |
|---|---|---|---|
| Average SNV distance | 53 (38–86) | 5 (1–24) | 3 (1–6) |
| Shared room[b] | 34 (1.0%) | 5 (3.8%) | 7 (4.2%) |
| Shared floor (any)[b] | 1,371 (39%) | 70 (54%) | 96 (58%) |
| Floor A | 102 (2.9%) | 6 (4.6%) | 6 (3.6%) |
| Floor B | 765 (22%) | 21 (16%) | 21 (13%) |
| Floor C | 252 (7.2%) | 10 (7.7%) | 7 (4.2%) |
| Floor D (includes high acuity unit) | 210 (6.0%) | 32 (25%) | 61 (37%) |
| Floor E | 42 (1.2%) | 1 (0.8%) | 1 (0.6%) |
| Average culture date difference (months) | 1.40 (0.57–3.27) | 0.93 (0.43–2.41) | 0.47 (0.20–1.37) |
| Antibiotic exposure in donor (any)[b] | 2,961 (85%) | 108 (83%) | 140 (85%) |
| Carbapenem antibiotic | 1,417 (40%) | 53 (41%) | 61 (37%) |
| Non-carbapenem antibiotic | 2,911 (83%) | 106 (82%) | 139 (84%) |
| Time period[c] | | | |
| Period 1 | 568 (16%) | 18 (14%) | 15 (9.1%) |
| Period 2 | 771 (22%) | 29 (22%) | 23 (14%) |
| Period 3 | 992 (28%) | 35 (27%) | 33 (20%) |
| Period 4 | 1,169 (33%) | 48 (37%) | 94 (57%) |

[a]Results are shown for (i) all pairs who overlapped in the facility, (ii) each recipient and their genomically closest possible donor (based on core genome SNV distance) who overlapped in the facility, and (iii) pairs who overlapped in the facility that also belong to the same genomic cluster. Medians with interquartile ranges are shown for continuous variables and counts with percentages are shown for categorical variables. CRKP, carbapenem-resistant *Klebsiella pneumoniae*; SNV, single nucleotide variant.
[b]During pairwise exposure period.
[c]Time period when recipient first tested positive for CRKP.

acuity unit) having the larger effect on pairwise genomic relatedness, suggesting there may be more intra-floor risk on floors where patients require more intensive care (Table 2). In Model 1 (pairwise SNV distance), both patients residing on Floor D were associated with SNV distances 44% (95% CI: 38–49%) closer. In Model 2 (closest donor), sharing time on Floor D was associated with 7 (95% CI: 4–13) times greater odds of being the closest potential donor. In Model 3 (same transmission cluster), sharing time on Floor D was associated with 10 (95% CI: 6–18) times greater odds of being in the same cluster. When collapsing the effect of floor sharing into a single covariate, sharing a floor during their exposure period was a significant predictor of genetic relatedness of CRKP isolate pairs in all models (Table S1).

Other factors, such as sharing a room during a pair's mutual exposure period, a shorter lag between positive cultures (Fig. S1), and the exposure period occurring in the fourth quarter of the study period was positively associated with isolate similarity in each model, although certainty varied. Model 3 identified all three of these factors as significantly related to cluster comembership, Model 1 captured two of these factors (shared room and time period) as significantly associated with SNV distances, and Model 2 failed to show a statistically significant effect of any of these factors on the likelihood of being the closest potential donor (Table 2). Antibiotic exposure of the donor during the pairwise exposure period was not a meaningful predictor of CRKP relatedness in any of the models.

Results were qualitatively similar when running our models with all pairs (including those that did not have overlapping stays in the facility) (Table S2), and sequential room sharing (at least 1 day between when the potential donor left a room and the recipient was admitted into the same room during the pairwise exposure period) did not seem to be associated with genetic relatedness of isolates (Table S3).

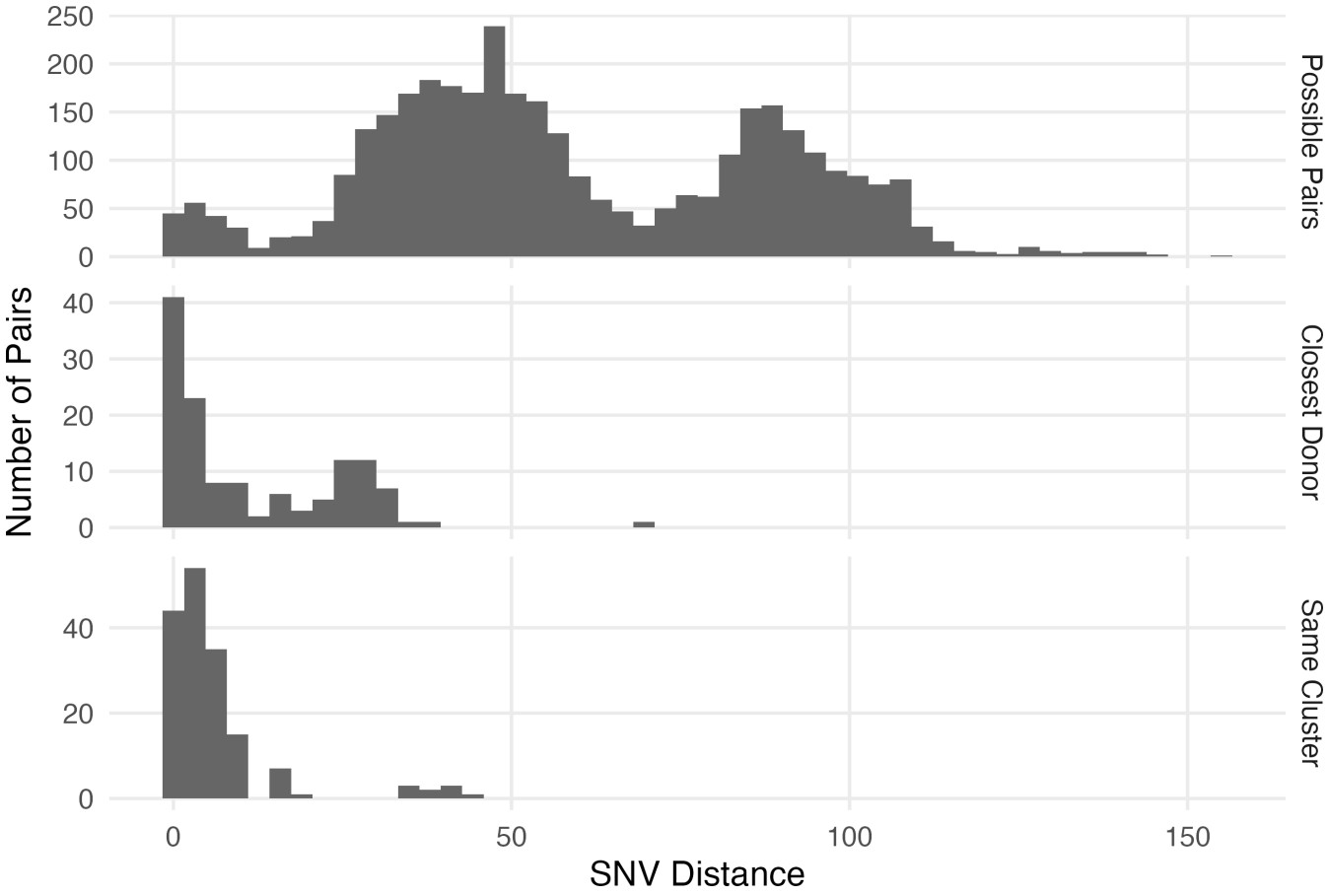

**FIG 1** Distribution of single nucleotide variant (SNV) distances between different types of infectious/exposed pairs. From top to bottom, the panels show the distribution of SNV distances for (i) pairs that overlapped in the facility, (ii) distances between each recipient and their genomically closest possible donor (based on core genome SNV distance) that overlapped in the facility, and (iii) pairs who overlapped in the facility that also belong to the same genomic cluster.

## Case and admission-positive only models underestimate key risk factors

Although our study utilized serial surveillance for asymptomatic carriage, most facilities have only clinical culture isolates available, with some also testing for CRKP coloniza- tion on admission. Of the 180 patients colonized with ST258 CRKP in our study, 30% would never have been identified if only clinical and admission screening cultures were conducted, and 61% of closest potential transmission pairs (defined as in Model 2) would have been missed. When we excluded these patients from our analyses, culture date difference remained a significant predictor of genomic similarity, but room and floor sharing was not significant in any of the three models (although the qualitative direction of coefficients remained unchanged) (Table S4).

## DISCUSSION

By using regression models to evaluate variables associated with genomic relatedness between pairs of isolates, we were able to identify risk factors for CRKP transmission in an endemic LTACH setting. Although certainty varied, regardless of the metric of genomic relatedness employed, sharing a room or having an overlapping stay on the same floor, especially the floor that included the high acuity unit, predicted shorter genomic distances and a higher likelihood of membership in the same cluster. This is consistent with studies showing increased risk of CRKP infection among those with more intense care needs (e.g. fecal incontinence and mechanical ventilation) and those exposed to infected roommates (22–24). This work suggests that decolonization and other infection

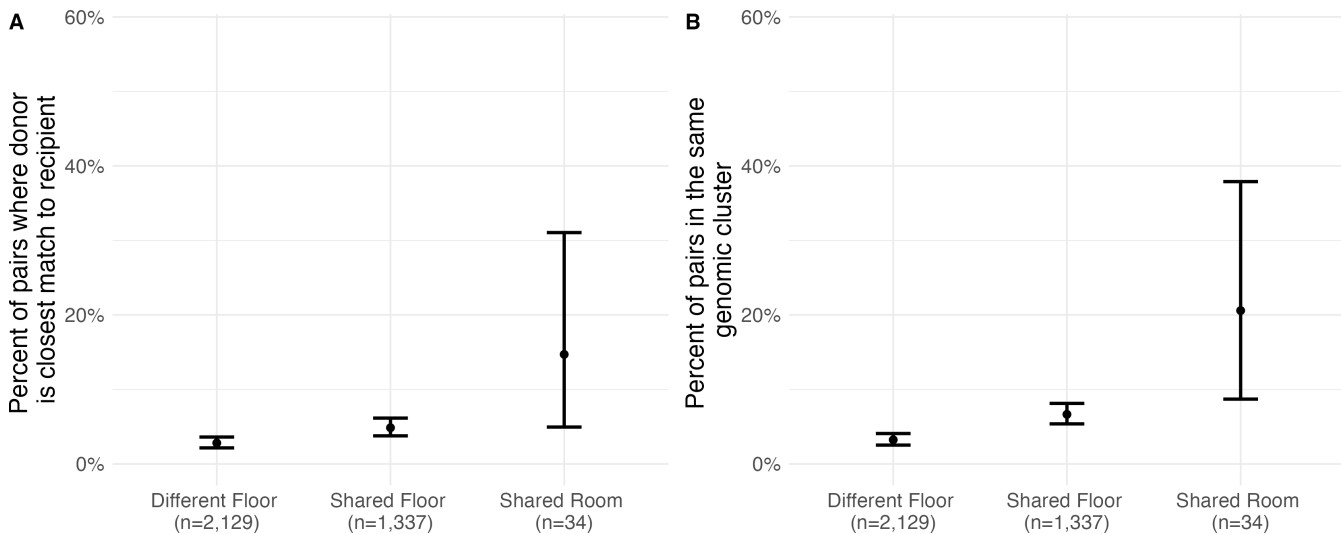

**FIG 2** Percent of potential transmission pairs where (A) the infectious individual is the closest potential donor for the recipient and (B) the infectious and exposed patients are in the same genomic cluster. Each panel shows risks associated with residing on different floors in the same facility, residing on the same floor, and from sharing a room. Vertical bars represent 95% confidence intervals.

prevention efforts should be focused on close within-facility contacts of CRKP patients, with particular attention to high-acuity patients who have higher illness severity and are likely to have more medical interventions and direct hands-on contact with staff.

The availability of WGS data from colonization isolates gave us the ability to evaluate how individual, dyadic, and contextual factors predicted the genetic similarity of CRKP

**TABLE 2** Drivers of variation in pairwise genomic relatedness as a function of individual and pair-level risk factors ($n$ = 174 patients, 3,500 pairs)[a]

| | Model 1. SNV distance model | Model 2. Closest donor model | Model 3. Same cluster model |
|---|---|---|---|
| Intercept | **51.65 (46.27–57.66)** | **0.02 (0.01–0.04)** | **0.01 (0.00–0.03)** |
| Shared Floor A[b] | **0.80 (0.70–0.91)** | **3.19 (1.21–8.42)** | **3.73 (1.32–10.54)** |
| Shared Floor B[b] | 1.00 (0.94–1.06) | 1.11 (0.63–1.95) | 1.22 (0.68–2.18) |
| Shared Floor C[b] | 1.08 (0.99–1.18) | 1.41 (0.66–3.03) | 1.08 (0.44–2.68) |
| Shared Floor D[b] (includes high acuity unit) | **0.56 (0.51–0.62)** | **7.12 (3.84–13.19)** | **10.1 (5.77–17.8)** |
| Shared Floor E[b] | **1.32 (1.07–1.62)** | 1.19 (0.14–10.28) | 0.45 (0.04–4.77) |
| Shared room[b] | **0.75 (0.60–0.93)** | 2.68 (0.78–9.17) | **4.15 (1.07–16.18)** |
| Culture date difference (30 days) | 1.01 (1.00–1.03) | 0.93 (0.82–1.06) | **0.77 (0.67–0.90)** |
| Carbapenem antibiotic exposure of donor[b] | 1.02 (0.96–1.08) | 1.16 (0.71–1.89) | 0.93 (0.55–1.59) |
| Non-carbapenem antibiotic exposure of donor[b] | 1.03 (0.96–1.11) | 0.72 (0.40–1.29) | 0.95 (0.51–1.79) |
| Time period (quarter 2 vs quarter 1)[c] | 0.95 (0.87–1.04) | 1.25 (0.61–2.59) | 1.60 (0.69–3.67) |
| Time period (quarter 3 vs quarter 1)[c] | 0.98 (0.88–1.10) | 0.97 (0.44–2.13) | 1.85 (0.78–4.42) |
| Time period (quarter 4 vs quarter 1)[c] | **0.87 (0.76–0.99)** | 1.23 (0.54–2.79) | **4.05 (1.66–9.85)** |

[a]Model 1 is a log-linear model of pairwise SNV distance as a function of individual and pairwise exposure risks. Coefficients are exponentiated and can be interpreted analogously to rate ratios, with values <1 indicating smaller distances and >1 indicating greater distances. Models 2 and 3 are logistic regression models characterizing changes in the odds that a given infectious case is the most closely related to the recipient (Model 2) or that the infectious case and exposed individual are in the same genomic cluster (Model 3). All results are adjusted for all covariates in the model, and a random effect for potential donor is included in the models. Estimates are exponentiated and 95% confidence intervals are in parentheses. Bolded values have confidence intervals that do not contain 1. SNV, single nucleotide variant.
[b]During pairwise exposure period.
[c]When recipient first tested positive.

isolates, a proxy for transmission risk, in a single LTACH from June 2012 to June 2013. Using these WGS data, we quantified the relatedness of CRKP isolate pairs in three ways: SNV distance, closest potential donor, and cluster co-membership. Each of these measures of relatedness yielded risk-factor estimates which were consistent with each other as well as existing literature on CRKP transmission. This suggests that the measure of relatedness used in pairwise models may be flexible. It may be important, however, to consider the sampling strategy and transmission dynamics of the pathogen of interest when selecting a pairwise metric. For instance, if serial sampling was not conducted and it is unlikely that direct transmission pairs have been identified, a closest donor approach may not be sensible. Or, if the pathogen of interest has a well-established SNV cutoff to determine cluster co-membership, using the criteria of a pair meeting that cutoff may be used as the model outcome. In our study population, it appears that a threshold-free cluster comembership model identifies transmission risk factors with the most certainty compared to a closest donor or SNV distance models.

Sensitivity analyses revealed that excluding data from serial surveillance isolates reduced our ability to identify the risk factors highlighted using the full data set. This likely reflected the decreased number of cases overall in the reduced data set as well as missed direct transmission links, highlighting the importance of serial culture surveillance as a tool for identifying transmission risk factors. When patients who did not have a clinical CRKP isolate during our study period and were negative on admission were excluded from analyses, 61% of probable transmission pairs were missed and our ability to detect an effect of room and floor sharing on genetic relatedness was weakened.

In addition to room and floor sharing, our models revealed that pairs are less likely to be closely genetically related if the time between collection of positive samples is longer. This suggests that a susceptible patient is more likely to get CRKP from someone who has more recently acquired CRKP than someone who has been colonized for a longer time. It could also indicate intra-host evolution between the time of acquisition and transmission. Two of our three models also suggested that individuals colonized during the last quarter of our study period were more closely related to their potential donors than those colonized during the first period. This could be an artifact of model setup (as the study period progressed, the number of potential donors increased), the result of the introduction of a new strain into the facility with different transmission patterns (which would be unlikely since all major strain types were present during the study period), or more intra-facility transmission in this period. However, the incidence of CRKP within the facility appeared to decrease throughout the study period (13), and thus this result may indicate the onward transmission of fewer strains within the facility resulting in those infected appearing to be more closely related to each other than if many strains were circulating due to a bottleneck effect. Future research can further explore this relationship between study period and genomic relatedness. Finally, although antibiotic exposure has been associated with CRKP acquisition risk in healthcare facilities (22, 23), our results do not provide evidence that antibiotic exposure is associated with a change in the number of transmissions generated by a colonized LTACH resident. However, the very high prevalence of antibiotic use in our patient population may have hindered the detection of its impact on transmission.

Due to data availability and methodological constraints, our study can be considered to have the following limitations. First, we did not have access to information on patient-level procedures, devices, and healthcare worker exposures, which all may play a role in transmission and could help determine specific mechanisms increasing transmission risks, specifically on Floor D. However, we were able to identify known risk factors of CRKP transmission in an LTACH and hope this study will serve as a template for facilities that may have more detailed data available. Additionally, given that only 58% of closest donor pairs were in the same genomic cluster, it is likely we are missing some direct transmission links of CRKP in this LTACH. Thus, even the closest donor model may not be completely representative of direct transmission between two patients, and this may be why some transmission risks were not identified in the closest donor model. However,

given our strategy of serially sampling every patient in the facility, there is a high probability of direct transmission links being represented in our model outcomes, so risk factors for direct transmission should be picked up even if not all transmission pairs are present in the data. This is supported by our results corresponding with known CRKP risk factors. Additionally, we are not only interested in direct transmission links but also transmission patterns of certain clusters of isolates, both of which could be identified in our models and helpful in infection prevention interventions. Finally, LTACH patients are frequently moved between different connected healthcare facilities for varying levels of care. We have previously shown the role of patient transfer in CRKP spread and the detection of closely related strains on admission of facilities (25, 26). Further research extending these models is necessary to better understand how shared exposures outside of the focal facility impact the genomic similarity of CRKP isolates observed at the facility level.

One caveat of our study design is that it includes demographic and contextual data on only CRKP positive patients. Thus, all inferences are conditional on both members of a pair being colonized. Accordingly, the epidemiologic risk factors identified in our results should be interpreted as driving genomic similarity between isolates from colonized individuals, as opposed to an individual's risk of colonization. This aspect of our study, however, makes it more accessible, as in many community settings, public health data sets only contain case data, and thus methods that necessitate uninfected controls would be infeasible.

As WGS pathogen data become more widely available and inexpensive to obtain, genomic data have an important role to play in routine surveillance of pathogens such as CRKP. Our analysis shows that these data can provide insights in high-prevalence settings which would not be accessible otherwise. Our results also underscore that the choice of genomic relatedness may be important but is also somewhat flexible and is likely to vary by context and goal of the surveillance activity. For example, infection prevention efforts targeted at mitigating the spread of novel drug-resistant variants may utilize different outcomes than those focused on identifying generic transmission risk factors of endemic pathogens. The approach outlined in this analysis requires few assumptions, including no arbitrary SNV cutoffs, no uninfected controls, and only modest computational power and suggests that routine WGS-based surveillance may allow for earlier detection and facility-specific intervention in nosocomial outbreaks of CRKP and other pathogens causing significant morbidity and mortality in vulnerable, hospitalized populations.

## ACKNOWLEDGMENTS

The authors thank Josh Warren for his consultation on genomic pairwise regression modeling for infectious disease transmission studies.

This research was supported by the National Institutes of Health (R01 AI148259). H.S., E.S., and J.Z. conceptualized the study and developed methodology. H.S. conducted analyses, created visualizations, and wrote the original draft. M.K.H. provided data and guidance in interpreting results. T.A. consulted throughout the project. All authors reviewed and edited this manuscript. Authors have no competing interests to report.

## AUTHOR AFFILIATIONS

[1]Department of Epidemiology, University of Michigan School of Public Health, Ann Arbor, Michigan, USA

[2]Center for Social Epidemiology and Population Health, University of Michigan School of Public Health, Ann Arbor, Michigan, USA

[3]Department of Microbiology and Immunology, University of Michigan, Ann Arbor, Michigan, USA

[4]Division of Infectious Diseases, Rush University Medical Center, Chicago, Illinois, USA

[5]Division of Infectious Diseases, Department of Medicine, University of Michigan, Ann Arbor, Michigan, USA

[6]University of Michigan Medical School, Ann Arbor, Michigan, USA

## AUTHOR ORCIDs

Hannah Steinberg http://orcid.org/0000-0003-3909-4158
Evan Snitkin http://orcid.org/0000-0001-8409-278X

## FUNDING

| Funder | Grant(s) | Author(s) |
| --- | --- | --- |
| National Institutes of Health | R01 AI148259 | Evan Snitkin |

## AUTHOR CONTRIBUTIONS

Hannah Steinberg, Conceptualization, Formal analysis, Investigation, Methodology, Visualization, Writing – original draft, Writing – review and editing | Timileyin Adediran, Methodology, Writing – review and editing | Mary K. Hayden, Data curation, Writing – review and editing | Evan Snitkin, Conceptualization, Investigation, Methodology, Supervision, Writing – review and editing | Jon Zelner, Conceptualization, Investigation, Methodology, Supervision, Writing – review and editing

## DATA AVAILABILITY

Sequence data and limited meta-data are available under National Center for Biotechnology Information BioProject PRJNA603790. Select analysis code available at https://github.com/hsteinberg/crkp_pairwise_regression_modeling_paper/.

## ADDITIONAL FILES

The following material is available online.

### Supplemental Material

**Supplemental tables and figure (Spectrum02452-25-s0001.pdf).** Tables S1 to S4 and Figure S1.

### Open Peer Review

**PEER REVIEW HISTORY (review-history.pdf).** An accounting of the reviewer comments and feedback.

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
