## [Reviewer comments · Microbiology Spectrum]

Microbiology Spectrum

Regression-based modeling of pairwise genomic linkage data identifies risk factors for healthcare-associated infection transmission: Application to carbapenem-resistant *Klebsiella pneumoniae* transmission in a long-term care facility

Hannah Steinberg, Timileyin Adediran, Mary Hayden, Evan Snitkin, and Jon Zelner

Corresponding Author(s): Evan Snitkin, University of Michigan Medical School

Review Timeline:

Submission Date:	August 10, 2025
Editorial Decision:	September 26, 2025
Revision Received:	December 2, 2025
Accepted:	December 8, 2025

Editor: Se-Ran Jun

Reviewer(s): The reviewers have opted to remain anonymous.

Transaction Report:

DOI: <https://doi.org/10.1128/spectrum.02452-25>

Re: Spectrum02452-25 (Regression-based modeling of pairwise genomic linkage data identifies risk factors for healthcare-associated infection transmission: Application to carbapenem-resistant *Klebsiella pneumoniae* transmission in a long-term care facility)

Dear Ms. Hannah Steinberg:

Thank you for the privilege of reviewing your work. Below you will find my comments, instructions from the Spectrum editorial office, and the reviewer comments.

Reviewers raised concerns about the subsetting of data included in this manuscript and robustness in statistics. Especially, the study only included patients whose visits to the hospital overlapped which could influence the outcome of models in predicting shared rooms/floors of the hospital as drivers of infection. It would be beneficial to see results obtained from considering all data and an explanation of why subsetting of data was made.

Revision Guidelines

Sincerely,
Se-Ran Jun
Editor
Microbiology Spectrum

Reviewer #1 (Public repository details (Required)):

WGS data

Reviewer #1 (Comments for the Author):

Significance

It is a very interesting and meaningful study to explore ways to decrease HAI.

1. Why was rectal swab collected, rather than oral saliva or skin swab? Rectal swab is much more difficult to obtain than the other two.

2. Within same cluster model, does it mean that

In the same cluster, samples own more shared characteristics and the intra-variation is less, so that it is easy to get the risk factor?

3. In table 2, what is the outcome variable? Apart from the variables listed in table 1, have considered any other potential variables influencing the WGS results, i.e., season, the volume of the unit, the type of disease etc.

4. Why are the three regression-based models (log-linear, two logistic regression) used? Does the data meet the assumptions of each model? Please show the results of testing the assumptions.

Reviewer #2 (Public repository details (Required)):

All whole genome sequence data should be submitted to NCBI and code used for analysis should be made publicly available on a repository after publication.

Reviewer #2 (Comments for the Author):

This study utilizes a large dataset from the long-term acute care hospital and three different regression-based models to identify risk factors for healthcare associated infection with carbapenem-resistance *Klebsiella pneumoniae*. The paper is generally well written and adds to a growing body of work that proposes the use of whole genome sequencing as a tool to prevent outbreaks of nosocomial infections during hospital stays.

General comments:

The use of "infection" and "colonization" are used early in the paper as distinct terms but are poorly defined. Later in the paper (particularly in the results) colonization and infection appear to be used interchangeably. Please amend this to clearly define CRKP colonization vs. infection and then use the terms consistently throughout the manuscript.

I think the manuscript could benefit from clearer aims and outcomes. It was unclear most of the way through if this is a methods paper (introducing WGS surveillance CRKP outbreak monitoring) or an observational study of an outbreak at a healthcare facility. I feel that the paper works best as a methods validation style manuscript but needs more information about the methods implemented (code used for analysis should be made publicly available on publication) and results should focus more on comparisons between methods than on the drivers of transmission.

Significance

It is a very interesting and meaningful study to explore ways to decrease HAI.

1. Why was rectal swab collected, rather than oral saliva or skin swab? Rectal swab is much more difficult to obtain than the other two.

2. Within same cluster model, does it mean that

In the same cluster, samples own more shared characteristics and the intra-variation is less, so that it is easy to get the risk factor?

3. In table 2, what is the outcome variable? Apart from the variables listed in table 1, have considered any other potential variables influencing the WGS results, i.e., season, the volume of the unit, the type of disease etc.

4. Why are the three regression-based models (log-linear, two logistic regression) used? Does the data meet the assumptions of each model? Please show the results of testing the assumptions.

Spectrum02452-25 Peer Review

This study utilizes a large dataset from the long-term acute care hospital and three different regression-based models to identify risk factors for healthcare associated infection with carbapenem-resistance *Klebsiella pneumoniae*. The paper is generally well written and adds to a growing body of work that proposes the use of whole genome sequencing as a tool to prevent outbreaks of nosocomial infections during hospital stays.

General comments:

The use of “infection” and “colonization” are used early in the paper as distinct terms but are poorly defined. Later in the paper (particularly in the results) colonization and infection appear to be used interchangeably. Please amend this to clearly define CRKP colonization vs. infection and then use the terms consistently throughout the manuscript.

I think the manuscript could benefit from clearer aims and outcomes. It was unclear most of the way through if this is a methods paper (introducing WGS surveillance CRKP outbreak monitoring) or an observational study of an outbreak at a healthcare facility. I feel that the paper works best as a methods validation style manuscript but needs more information about the methods implemented (code used for analysis should be made publicly available on publication) and results should focus more on comparisons between methods than on the drivers of transmission.

Line 83: is there a more up-to-date citation for this figure about number of patients on a given day with healthcare associated infections?

Line 85: “Antibiotic resistance is common in healthcare pathogens” please provide citation

Lines 102 – 103: Please comment on the role of horizontal gene transfer in HAIs, either broadly or specifically in relation to *Klebsiella*.

Line 106: definition should precede first use of acronym (SNV)

Line 114: Please expand on the phrase “drivers of transmission” – are you seeking to identify human-based drivers (i.e. shared healthcare workers, contaminated equipment) or genetic drivers of increased transmissibility in the pathogen itself? If you are not seeking information from the pathogen genome itself, please include some detail about the potential uses of your WGS data in the future for this purpose.

Line 121: Include more information about the types of infection observed in this study. All samples were collected via rectal swab – is this always an appropriate sampling method if the patient had pneumonia or a UTI? If this is standard of care sampling for CRKP, please state that in the text.

Lines 143 – 144: Please include the study ID in the text.

Line 146: “Surveillance samples were cultures and unique colony morphologies were identified to species.” This sentence doesn’t make sense. Do you mean “per species”? Or “to species-level”?

Line 147: Please change to “Ertapenem disks were used to screen isolates for carbapenem resistance”

Line 148: Define CRE

Lines 148 – 149: Please include a citation (or method) for the carbapenemase gene present in the region during the study period

Lines 150 – 151: “Whole genome sequences were obtained for all positive isolates” – please expand for clarity. Positive by both disk assay and PCR?

Lines 152 – 153: Please include the total number of isolates used in the final analysis.

Lines 255 – 257: This sentence does not make sense.

Line 296: Please include the range of times for which patients were infected

Line 301: You mention the possibility of a new strain introduction into the facility – can you determine this from your WGS data? This would be an important note if yes, and if not should be mentioned as a drawback of the study/sequencing data collected.

Lines 330 – 333: Previous study also did not test equipment in shared rooms, could be missing an important transmission source. As the strains are all relatively closely related it seems unlikely that patients are consistently coming into the facility with entirely new strains and they are more than likely picking up strains at the facility. It seems an odd choice not to include these data and may be skewing outcomes in the favor of room/floor sharing as drivers of transmission.

Lines 337 – 339: Please ensure that this caveat is also made clear in the introduction.

Figure 2: It would be useful to see the distribution of datapoints on this plot, either by dot plot or violin plot overlaid with CI error bars.

Reviewer 1

Significance

It is a very interesting and meaningful study to explore ways to decrease HAI.

1. Why was rectal swab collected, rather than oral saliva or skin swab? Rectal swab is much more difficult to obtain than the other two.

Rectal swabs are used for detection of CRKP carriage due to the gut being the primary site of colonization. Thus, for CRKP active surveillance programs in healthcare settings, this is standard practice.

2. Within same cluster model, does it mean that in the same cluster, samples own more shared characteristics and the intra-variation is less, so that it is easy to get the risk factor?

In the first paragraph of the results we explain that clusters were defined by “a threshold-free cluster detection approach that clustered each CRKP isolate acquired at the LTACH to the importation isolate with which it shared the greatest number of variants”. More information is available in Hawken et al., Lancet Microbe, 2022, which is cited.

3. In table 2, what is the outcome variable? Apart from the variables listed in table 1, have considered any other potential variables influencing the WGS results, i.e., season, the volume of the unit, the type of disease etc.

Outcome variables are listed in the table description:

“Model 1 is a log-linear model of pairwise SNV distance as a function of individual and pairwise exposure risks. Coefficients are exponentiated and can be interpreted analogously to rate ratios, with values < 1 indicating smaller distances and > 1 indicating greater distances. Models 2 & 3 are logistic regression models characterizing changes in the odds that a given infectious case is the most closely related to the recipient (Model 2) or that the infectious case and exposed individual are in the same genomic cluster (Model 3). All results are adjusted for all covariates in the model, and a random effect for potential donor is included in the models.”

We included study period in the models, and all isolates were from colonization surveillance.

4. Why are the three regression-based models (log-linear, two logistic regression) used? Does the data meet the assumptions of each model? Please show the results of testing the assumptions.

Each model type was chosen based on the structure of the outcome variable being evaluated (log-linear for SNV distance and logistic for the binary variables of closest donor and same cluster).

We adjusted for the nonindependence of our pairwise observations by including random intercepts that account for unobserved factors that may make an individual more or less influential than would be suggested by our measured covariates. To clarify this, we added this sentence in the methods:

“To account for the influence of unusually infectious individuals and nonindependence of pairs with shared cases, all models included a random intercept term for each potential donor.”

To ensure that our results are robust to non-linear associations between covariates and pairwise relatedness, we utilized categorical predictors for all variables other than culture date difference. In addition, to ensure that our model was well-specified for culture date difference, we included new Supplemental Figure 1, reproduced below, to verify that that this assumption is met for all models.

Supplemental Figure 1. Relationship between pairwise culture date difference and model outcomes. Culture date difference appears to **(A)** not be associated with log mean single nucleotide variant (SNV) distance, and to have a log linear negative relationship with **(B)** pairs in the same cluster and **(C)** pairs that include the recipient's closest donor. This satisfies the assumptions of log-linear relationships between the linear predictor and outcome in our pairwise regression models. Due to low sample size, culture date differences of ten or more months are categorized as ten months (maximum culture date difference = 12 months).

Reviewer 2

This study utilizes a large dataset from the long-term acute care hospital and three different regression-based models to identify risk factors for healthcare associated infection with carbapenem-resistance *Klebsiella pneumoniae*. The paper is generally well written and adds to a growing body of work that proposes the use of whole genome sequencing as a tool to prevent outbreaks of nosocomial infections during hospital stays.

General comments:

The use of “infection” and “colonization” are used early in the paper as distinct terms but are poorly defined. Later in the paper (particularly in the results) colonization and infection appear to be used interchangeably. Please amend this to clearly define CRKP colonization vs. infection and then use the terms consistently throughout the manuscript.

Thank you for this comment. We have carefully edited the paper to ensure that these terms are used appropriately throughout the manuscript.

I think the manuscript could benefit from clearer aims and outcomes. It was unclear most of the way through if this is a methods paper (introducing WGS surveillance CRKP outbreak monitoring) or an observational study of an outbreak at a healthcare facility. I feel that the paper works best as a methods validation style manuscript but needs more information about the methods implemented (code used for analysis should be made publicly available on publication) and results should focus more on comparisons between methods than on the drivers of transmission.

To address this concern, we have modified the first sentence of the second to last paragraph of the introduction to read as follows:

*“In this analysis, we aim to further validate the use of pairwise regression models incorporating WGS and epidemiologic data to study infectious disease transmission in endemic settings by evaluating the use of these models to describe how carbapenem-resistant *Klebsiella pneumoniae* (CRKP), an important healthcare-associated pathogen, is transmitted in a long-term acute care hospital (LTACH) using isolates collected on a regular basis as part of a surveillance study¹⁰.”*

We believe that further clarifies the purpose of our study as a validation of this new method.

We have also made our analysis code available here:

https://github.com/hsteinberg/crkp_pairwise_regression_modeling_paper/. We are however unable to provide the corresponding data for this project because it is classified as protected health information. We also added more information to the methods section of the manuscript about the R packages we used to the methods:

- “a log-linear model of core genome single nucleotide variant (SNV) distances between the two isolates (calculated raw SNV distances with the *dist.dna* function in the *ape* R package v5.8-1¹⁴”
 - “all models were run using the *gImer* function in the *lme4* package v1.1-35.3.”
- We hope this addition along with the detailed description of our models allow others to replicate our methods with their own data.

Line 83: is there a more up-to-date citation for this figure about number of patients on a given day with healthcare associated infections?

Through a brief literature search, there does not appear to be an updated citation for this statistic. The CDC still cites this figure (<https://www.cdc.gov/healthcare-associated-infections/php/data/index.html>).

Line 85: “Antibiotic resistance is common in healthcare pathogens” please provide citation

Added two citations:

Edelsberg, J. *et al.* Prevalence of antibiotic resistance in US hospitals. *Diagn. Microbiol. Infect. Dis.* **78**, 255–262 (2014).

GBD 2021 Antimicrobial Resistance Collaborators. Global burden of bacterial antimicrobial resistance 1990–2021: a systematic analysis with forecasts to 2050. *Lancet Lond. Engl.* **404**, 1199–1226 (2024).

Lines 102 – 103: Please comment on the role of horizontal gene transfer in HAIs, either broadly or specifically in relation to *Klebsiella*.

Added: While this has been done to some degree in outbreak settings,^{6,7} there is still a need for methods applicable in high-prevalence endemic settings, where the constant importation of resistant organisms **and frequent horizontal gene transfer of resistance-conferring genetic elements**⁸ makes delineating transmission links challenging, even with genomic data.

Line 106: definition should precede first use of acronym (SNV)

Fixed.

Line 114: Please expand on the phrase “drivers of transmission” – are you seeking to identify human-based drivers (i.e. shared healthcare workers, contaminated equipment) or genetic drivers of increased transmissibility in the pathogen itself? If you are not seeking information from the pathogen genome itself, please include some detail about the potential uses of your WGS data in the future for this purpose.

Changed to “epidemiologic drivers of transmission”.

Line 121: Include more information about the types of infection observed in this study. All samples were collected via rectal swab – is this always an appropriate sampling

method if the patient had pneumonia or a UTI? If this is standard of care sampling for CRKP, please state that in the text.

Added: “using isolates collected on a regular basis as part of a surveillance study” to indicate that these were surveillance isolates collected as part of a prospective surveillance research study.

Lines 143 – 144: Please include the study ID in the text.

Added study ID HUM00174826 to text.

Line 146: “Surveillance samples were cultures and unique colony morphologies were identified to species.” This sentence doesn’t make sense. Do you mean “per species”? Or “to species-level”?

Thanks for pointing this out. Changed to “species-level.”

Line 147: Please change to “Ertapenem disks were used to screen isolates for carbapenem resistance”

Changed.

Line 148: Define CRE

Done.

Lines 148 – 149: Please include a citation (or method) for the carbapenemase gene present in the region during the study period

Added reference

Lin, M. Y. *et al.* The Importance of Long-term Acute Care Hospitals in the Regional Epidemiology of *Klebsiella pneumoniae* Carbapenemase–Producing Enterobacteriaceae. *Clin. Infect. Dis. Off. Publ. Infect. Dis. Soc. Am.* **57**, 1246–1252 (2013).

Lines 150 – 151: “Whole genome sequences were obtained for all positive isolates” – please expand for clarity. Positive by both disk assay and PCR?

Clarified these isolates were positive on both entrapenem disk and *bla*_{KPC} PCR assays.

Lines 152 – 153: Please include the total number of isolates used in the final analysis.

This information is available in the *Prevalence of colonization and infection with CRKP in a single LTACH section* of the Results: “In total, 255 individuals were colonized with at least one strain of CRKP during the study period (with an average prevalence of 32% throughout the year),¹³ 180 of whom (70% of those colonized) were colonized with strain ST258.”

Lines 255 – 257: This sentence does not make sense.

We have edited this sentence as follows to ensure it reads clearly: “By using regression models to evaluate variables associated with genomic relatedness between pairs of isolates, we were able to identify risk factors for CRKP transmission in an endemic LTACH setting.”

Line 296: Please include the range of times for which patients were infected

Added “in a single LTACH from June 2012-June 2013”

Line 301: You mention the possibility of a new strain introduction into the facility – can you determine this from your WGS data? This would be an important note if yes, and if not should be mentioned as a drawback of the study/sequencing data collected.

Thank you for this comment. We added the following to the discussion to address this concern:

“Future research can further explore this relationship between study period and genomic relatedness.”

We believe this model is important for hypothesis generating and additional research could parse out this relationship but as the purpose of this paper is to provide an example of the use of pairwise models for making sense of genomic and epidemiologic data in a long-term care facility, we do not think that exploring strain selection would benefit this particular manuscript.

We also added that a new strain “would be unlikely since all major strain types were present during the study period.”

Lines 330 – 333: Previous study also did not test equipment in shared rooms, could be missing an important transmission source. As the strains are all relatively closely related it seems unlikely that patients are consistently coming into the facility with entirely new strains and they are more than likely picking up strains at the facility. It seems an odd choice not to include these data and may be skewing outcomes in the favor of room/floor sharing as drivers of transmission.

This would be a great variable to add to these types of models, but unfortunately, as mentioned in the discussion, we did not have data on shared equipment in this facility. We did however conduct an additional analysis that included sequential room sharing and found this to not influence model outcomes (see new Supplemental Table 4, reproduced below).

Also, to address the comment about closely related strains entering the facility- patients frequently moved between different healthcare facilities in the area. We therefore hypothesize that closely related strains detected on admission are due to transmission occurring at connected healthcare facilities. This hypothesis is supported by the known role of patient transfer in CRKP spread, and the detection of closely related strains on admission. We added the following sentences to the discussion: “Finally, LTACH patients are frequently moved between different connected healthcare facilities for varying levels of care. We have previously shown the role of patient transfer in CRKP spread and the detection of closely related strains on admission of facilities^{25,26}. Further research could extend these models to look at shared transferring facilities on genomic similarity of CRKP isolates.”

Supplemental Table 4. Drivers of variation in pairwise genomic relatedness as a function of individual and pair-level risk factors including pairs that did not overlap in the facility and successive room sharing variables (n = 180 patients, 10,931 pairs). Model 1 is a log-linear model of pairwise SNV distance as a function of individual and pairwise exposure risks. Coefficients are exponentiated and can be interpreted analogously to rate ratios, with values < 1 indicating smaller distances and > 1 indicating greater distances. Models 2 & 3 are logistic regression models characterizing changes in the odds that a given infectious case is the most closely related to the recipient (Model 2) or that the infectious case and exposed individual are in the same genomic cluster (Model 3). All results are adjusted for all covariates in the model, and a random effect for potential donor is included in the models.

	Model 1. SNV Distance Model	Model 2. Closest Donor Model	Model 3. Same Cluster Model
Intercept	52.22 (45.67, 59.71)	0.01 (0.00, 0.02)	0.00 (0.00, 0.01)
Shared Floor A ¹	0.81 (0.73, 0.89)	5.11 (1.76, 14.83)	3.19 (1.15, 8.83)
Shared Floor B ¹	0.98 (0.94, 1.02)	1.86 (1.02, 3.38)	1.40 (0.82, 2.37)
Shared Floor C ¹	1.04 (0.97, 1.12)	1.62 (0.63, 4.17)	1.46 (0.59, 3.60)
Shared Floor D ¹ (Includes High Acuity Unit)	0.59 (0.55, 0.64)	9.94 (5.16, 19.15)	9.21 (5.50, 15.43)
Shared Floor E ¹	1.31 (1.11, 1.54)	2.27 (0.25, 20.37)	0.45 (0.04, 4.89)
Shared room			
During pairwise exposure period	0.72 (0.60, 0.86)	4.47 (1.15, 17.29)	4.00 (0.94, 16.94)
<7 days before pairwise exposure period	1.11 (0.96, 1.28)	0.93 (0.11, 7.63)	1.08 (0.14, 8.62)
7-30 days before pairwise exposure period	1.06 (0.96, 1.16)	0.48 (0.10, 2.25)	0.86 (0.24, 3.05)
31-90 days before pairwise exposure period	1.01 (0.94, 1.09)	0.63 (0.17, 2.34)	1.75 (0.74, 4.10)
91 days-1 year before pairwise exposure period	0.99 (0.94, 1.04)	1.33 (0.51, 3.47)	1.82 (0.79, 4.20)
Culture date difference (30 days)	1.01 (1.00, 1.02)	0.79 (0.71, 0.87)	0.65 (0.58, 0.72)
Carbapenem antibiotic exposure of donor ²	0.96 (0.90, 1.02)	1.31 (0.73, 2.37)	1.58 (0.88, 2.83)
Non-Carbapenem antibiotic exposure of	1.00 (0.88, 1.13)	0.84 (0.29, 2.45)	1.94 (0.61, 6.19)

donor ²			
Time period (quarter 2 v quarter 1) ³	0.95 (0.90, 1.00)	1.32 (0.65, 2.67)	3.08 (1.52, 6.22)
Time period (quarter 3 v quarter 1) ³	1.03 (0.96, 1.10)	1.06 (0.49, 2.29)	3.61 (1.64, 7.92)
Time period (quarter 4 v quarter 1) ³	0.92 (0.84, 1.00)	1.58 (0.68, 3.67)	5.98 (2.52, 14.19)

Estimates are exponentiated and 95% confidence intervals are in parentheses. Bolded values have confidence intervals that do not contain 1.

¹During pairwise exposure period

²During period between when the donor last tested negative and the recipient tested positive

³When recipient first tested positive

SNV = single nucleotide variant.

Lines 337 – 339: Please ensure that this caveat is also made clear in the introduction. Added “This method utilizes WGS data from infected cases” to the last sentence of the introduction.

Figure 2: It would be useful to see the distribution of datapoints on this plot, either by dot plot or violin plot overlaid with CI error bars.

As the figure represents proportions, all datapoints are either 0 or 1, so we added sample sizes instead to better show what is being represented.

Editor

Reviewers raised concerns about the subsetting of data included in this manuscript and robustness in statistics. Especially, the study only included patients whose visits to the hospital overlapped which could influence the outcome of models in predicting shared rooms/floors of the hospital as drivers of infection. It would be beneficial to see results obtained from considering all data and an explanation of why subsetting of data was made.

We have added a new supplemental table (Supplemental Table 3, reproduced below) showing results without subsetting the data to patients that overlapped in the facility. Results were qualitatively unchanged. We also included a supplemental table (Supplemental Table 4, see above) that included successive room sharing in our models, which did not appear to influence pairwise genomic linkage.

Supplemental Table 3. Drivers of variation in pairwise genomic relatedness as a function of individual and pair-level risk factors including pairs that did not overlap in the facility (n = 180 patients, 10,931 pairs). Model 1 is a log-linear model of pairwise SNV distance as a function of individual and pairwise exposure risks. Coefficients are exponentiated and can be interpreted analogously to rate ratios, with values < 1 indicating smaller distances and > 1 indicating greater distances. Models 2 & 3 are logistic regression models characterizing changes in the odds that a given infectious case is the most closely related to the recipient (Model 2) or that the infectious case and exposed individual are in the same genomic cluster (Model 3). All results are adjusted for all covariates in the model, and a random effect for potential donor is included in the models.

	Model 1. SNV Distance Model	Model 2. Closest Donor Model	Model 3. Same Cluster Model
Intercept	52.27 (45.72, 59.76)	0.01 (0.00, 0.02)	0.00 (0.00, 0.01)
Shared Floor A ¹	0.81 (0.74, 0.90)	4.53 (1.58, 13.03)	3.36 (1.23, 9.20)
Shared Floor B ¹	0.99 (0.95, 1.03)	1.79 (0.99, 3.25)	1.41 (0.83, 2.38)
Shared Floor C ¹	1.05 (0.97, 1.12)	1.58 (0.62, 4.04)	1.46 (0.59, 3.59)
Shared Floor D ¹ (Includes High Acuity Unit)	0.59 (0.55, 0.64)	9.76 (5.09, 18.74)	9.27 (5.54, 15.52)
Shared Floor E ¹	1.31 (1.11, 1.54)	2.27 (0.25, 20.32)	0.46 (0.04, 5.00)
Shared room ¹	0.72 (0.60, 0.86)	4.57 (1.18, 17.63)	3.86 (0.91, 16.30)
Culture date difference (30 days)	1.01 (1.00, 1.02)	0.79 (0.72, 0.87)	0.65 (0.59, 0.72)
Carbapenem antibiotic exposure of donor ²	0.96 (0.90, 1.02)	1.30 (0.72, 2.34)	1.59 (0.89, 2.85)
Non-Carbapenem antibiotic exposure of	1.00 (0.88, 1.13)	0.84 (0.29, 2.43)	1.93 (0.60, 6.16)

donor ²			
Time period (quarter 2 v quarter 1) ³	0.95 (0.90, 1.00)	1.30 (0.65, 2.62)	3.20 (1.58, 6.46)
Time period (quarter 3 v quarter 1) ³	1.03 (0.96, 1.10)	1.06 (0.49, 2.28)	3.71 (1.69, 8.15)
Time period (quarter 4 v quarter 1) ³	0.92 (0.84, 1.00)	1.61 (0.70, 3.71)	6.22 (2.62, 14.75)

Estimates are exponentiated and 95% confidence intervals are in parentheses. Bolded values have confidence intervals that do not contain 1.

¹During pairwise exposure period

²During period between when the donor last tested negative and the recipient tested positive

³When recipient first tested positive

SNV = single nucleotide variant.

Re: Spectrum02452-25R1 (Regression-based modeling of pairwise genomic linkage data identifies risk factors for healthcare-associated infection transmission: Application to carbapenem-resistant *Klebsiella pneumoniae* transmission in a long-term care facility)

Dear Dr. Evan S Snitkin:

Your manuscript has been accepted, and I am forwarding it to the ASM production staff for publication. Your paper will first be checked to make sure all elements meet the technical requirements. ASM staff will contact you if anything needs to be revised before copyediting and production can begin. Otherwise, you will be notified when your proofs are ready to be viewed.

Sincerely,
Se-Ran Jun
Editor
Microbiology Spectrum